# Borderline Personality Symptoms, Body Modification, and Emotional Regulation

**DOI:** 10.3390/ijerph22010089

**Published:** 2025-01-10

**Authors:** Victoria Avon, Nathalie Gullo, D. Catherine Walker

**Affiliations:** 1Department of Psychology, Union College, Schenectady, NY 12308, USA; victoria.avon.321@gmail.com; 2Department of Psychological & Brain Sciences, Washington University, St. Louis, MO 63130, USA; gullonathalie@gmail.com

**Keywords:** borderline personality disorder, body modification, tattoos and piercings, emotional regulation, non-suicidal self-injury

## Abstract

Many people with BPD (borderline personality disorder) experience emotional dysregulation and thus engage in NSSI (non-suicidal self-injury), potentially in the pursuit of emotional regulation. However, research is lacking on whether body modifications (piercings, tattoos, etc.) are linked to BPD in a similar way to NSSI. In the current study, we hypothesized (1) that body modifications are associated with BPD symptoms, (2) that emotional regulation and self-expression motivations for body modifications specifically account for variance in BPD symptoms, and (3) that NSSI craving correlates with body modification craving. Participants (*N* = 199, ages 18–67, located in the USA) were surveyed on BPD symptomatology, NSSI craving, emotional regulation abilities, and the presence of body modifications. The extent of tattooing (number of tattoos and percentage of body surface covered) was not significantly associated with BPD symptomatology, but the number of piercings was. Individuals with higher BPD symptomatology were not more likely to report emotional regulation and self-expression as motivations for obtaining body modifications. NSSI craving scores were significantly positively correlated with body modification craving scores. Body modification may be an alternative method of emotional regulation to NSSI in individuals with BPD, which clinicians may want to consider when treating those with BPD and NSSI.

## 1. Introduction

About 1.6% of the global population is estimated to meet the diagnostic criteria for borderline personality disorder (BPD) [1]. The Diagnostic and Statistical Manual of Mental Disorders-5-TR (DSM-5-TR) defines BPD as “a pervasive pattern of instability of interpersonal relationships, self-image, and affects, and marked impulsivity, beginning by early adulthood and present in a variety of contexts” [1]. Diagnostic criteria for BPD include desperate efforts to avoid abandonment, volatile interpersonal relationships, identity disturbance, impulsivity, suicidal ideation and behavior, mood instability, feelings of chronic emptiness, intense anger, paranoid ideation, and non-suicidal self-injury (NSSI) [1]. In particular, BPD is a disorder characterized by emotional dysregulation, which may lead individuals to seek any release from overwhelming feelings, even if that release is maladaptive in the long term. Clinicians are advised to look for cutting scars and other evidence of NSSI when evaluating a patient with BPD [2]. Treatments for BPD typically focus on distress tolerance, mindfulness, emotional regulation, and interpersonal skills [2].

The phenomenon of NSSI involves intentional harm to oneself without suicidal motivations. While NSSI occurs in the general population and with other psychiatric disorders (estimated lifetime prevalence = 22.0%) [3], its prevalence is particularly elevated in BPD [4], with prevalence estimates of 65–80% in BPD samples [5,6]. Furthermore, many patients report “cravings” to engage in NSSI, similar to cravings seen in addictions [7]. The notable difference between cravings for substances and NSSI is that NSSI is craved primarily as a response to negative emotions rather than also as a means of eliciting positive emotions [8]. Nonetheless, other research has supported addictive-like features of NSSI [9]. In addition to emotional regulation, motives to engage in NSSI include self-derogation and interpersonal functions [6]. To assess whether NSSI provided temporary relief from negative emotions for those with personality disorders such as BPD, Snir et al. [10] prompted participants with BPD and avoidant personality disorder (APD) at five random times throughout the day to complete an experience-sampling digital diary, in which they described their affective, interpersonal, and behavioral (including NSSI) experiences at the current moment. With both BPD and APD patients, feelings of perceived rejection/isolation increased prior to engaging in NSSI and decreased afterward. Although participants with BPD and APD endorsed using NSSI for relief from unwanted emotions, participants with BPD and APD reported no significant change in general negative affect after engaging in NSSI [10]. This may speak to the function of NSSI across various personality disorders. Furthermore, in individuals who had previously engaged in NSSI and people with BPD, physiological arousal decreased during and after listening to an audio clip on the topic of NSSI [11,12]. Thus, people may engage in NSSI to cope with emotional dysregulation, perceiving that it aids in emotion regulation, whether it succeeds in doing so.

It may be possible that some with BPD might elect to get body modifications (i.e., tattoos or piercings) for a similar emotional regulation purpose. Studies have investigated the link between symptoms of BPD and body modifications in clinical and non-clinical populations. When surveying adolescents in a socio-rehab center, individuals with a greater number of body modifications were more likely to engage in alcohol/drug use, aggressive behavior, and self-destructive behavior [13], similar to BPD symptoms of impulsivity and difficulty controlling anger [1]. Wohlrab et al. [14] listed broad categories that may explain motivations for obtaining body modifications in a non-clinical sample. These categories include beauty/art/fashion, individuality, personal narrative, physical endurance, group affiliations/commitment, resistance, spirituality/cultural tradition, addiction, sexual motivation, and no specific reason. Although not clearly stated, some of these categories may be broadly extrapolated to underlying motives for NSSI and may relate to BPD symptoms. For example, individuality may relate to the concept of self-expression. Since many people with BPD experience an unstable and unclear self-concept, permanently altering the body in a way that reflects one’s uniqueness may combat discomfort from an unclear self-concept. The personal narrative motive may also be considered a form of emotional regulation; ‘reclaiming’ a part of one’s body that was once violated may aid in sustaining a more positive emotional outlook. Those with BPD, however, may experience different motives for body modification than in non-clinical samples, such as emotional regulation. Emotional regulation [15] and body modification [16] were investigated in victims of childhood abuse, many of whom have BPD [15]. Participants who had reported childhood abuse/neglect on a survey were more likely to report body modification, and participants who had endorsed more severe childhood abuse/neglect reported more piercings and tattoos than those with less severe abuse/neglect [16]. Although Ernst et al. [16] did not assess individual motivations for body modification, researchers suggested that coping with prior adversity and expressing autonomy may be a motivation for body modification. Similarly, physical endurance motivation refers to being able to withstand the pain of the body modification process [14], which may be used in a manner similar to NSSI. Lastly, addiction refers to the potential addictive component of getting new body modifications [14], which may relate to proneness to addictive behaviors seen among those with BPD [17] or cravings for engaging in NSSI [8].

Body modifications, NSSI, and emotional regulation may be uniquely interconnected in people with BPD. Blay et al. [4] evaluated BPD patients’ symptoms, symptom severity, NSSI, and body modifications. Approximately 69.8% of BPD patients had at least one tattoo [4], compared to a general population rate of around 27.3% [18]. Furthermore, the number of piercings an individual had was significantly positively correlated with their NSSI scores [4]. Similarly, among participants recruited through Craigslist advertisements, there was a positive correlation between the total number of body modifications a participant had (piercing, tattooing, scarification, etc.) and their degree of BPD symptoms [19]. Finally, some who engage in NSSI obtain tattoos to cover up scars or to prevent further NSSI [20], as they may not want to scar their tattoos.

Body modifications may even be an explicit alternative to NSSI or a protective factor against NSSI. In a study by Claes et al. [21], patients with eating disorders were asked to complete a self-report questionnaire, including assessments of self-injury habits and tattoos and body piercings. A negative correlation was found between body modifications and self-injury, and patients with piercings exhibited less severe eating disorder symptoms [16]. Ernst et al. [16] and Claes et al. [21] suggest that body modifications may serve as a socially acceptable form of altering the body for the sake of regulating aversive emotional states. Among participants subscribing to a tattooing and body modification magazine, those who reported cutting themselves had a greater number of piercings than those who did not cut themselves, were more likely to report being addicted to body modifications, and more frequently reported the expected physical pain as motivation for getting body modifications [22]. However, this study did not evaluate whether there was a linear relationship between the number and surface area of body modifications and the amount or frequency of NSSI. Consistent with the proposal that body modifications may serve as a protective factor against NSSI [21], participants recruited from internet piercing communities, in which 50% of participants reported a history of NSSI, stated that NSSI decreased after obtaining piercings, and 25% reported that their self-harm ceased altogether [23]. Thus, body modifications may replace or correlate with decreased self-injury over time.

The concept of body modifications as a method for emotional regulation and an alternative to NSSI is exemplified in a case study by Anderson and Sansone [24]. The case details a 19-year-old man with depression who used tattooing to regulate undesirable emotional states and to ward off thoughts of self-harm. He reported that the physical pain of tattooing provided a temporary distraction from emotional pain that would ‘take [his] mind off it.’ The young man asserted that scars from cutting would have been considered embarrassing, whereas his tattoos were considered “cool”. He also explained that more intense emotional pain would lead him to choose a more sensitive area for tattoo placement [24]. Thus, it is clear that emotional regulation, NSSI, and body modifications relate; however, it remains unclear whether body modifications are being used as a more adaptive or socially acceptable emotional regulation strategy for those who would otherwise engage in NSSI.

The current study aimed to extend the work of Blay et al. [4]. Rather than recruiting individuals diagnosed with BPD, the current study investigated BPD symptom severity in the general population. Specifically, we hypothesized that (H1) NSSI craving would be significantly associated with body modifications craving scores among participants who had body modifications; (H2) body modifications (e.g., number of tattoos/piercings, percentage of tattoo body coverage, locations of tattoos/piercings) would account for significant variance in BPD symptom scores; and (H3) self-expression and emotional regulation motivations for body modifications would account for significant variance in BPD symptom scores.

## 2. Materials and Methods

### 2.1. Participants

Participants (*N* = 199) were recruited through the researchers’ departmental subject pool and flyers distributed across college campuses, posted on social media, and posted in local tattoo parlors and hair salons in a mid-sized metropolitan area in the Northeast U.S. to survey a wide variety of individuals. This recruitment strategy was selected to maximize variability within the sample in terms of age and presence/absence of body modifications, which should improve the generalizability of the results. We were specifically targeting individuals who would have higher numbers of body modifications by recruiting from tattoo and piercing parlors in person and via their social media platforms and recruiting in hair salons to balance the sample with individuals who may be less likely to have body modifications. Individuals 18 and older were eligible to participate in the study and were able to enter a raffle for one of three cash prizes (USD 200, USD 100, and USD 50) following participation. The sample size was determined by the time frame available for completion of this study by the internal funding mechanism, attempting to recruit as many participants as possible within that time frame for adequate statistical power. Responses were collected from October 2023 to January 2024. Participants’ *M*(*SD*)_AGE_ was 26.87 (10.74) years old, ranging from 18–66. Participants’ demographic information is presented in Table 1 (due to researcher error, demographics were not collected for the first 95 participants. Demographic information is presented for the remaining 104 participants).

### 2.2. Measures

#### 2.2.1. Borderline Symptom List

The Borderline Symptom List (BSL) [25] is a 23-item, Likert-type scale with response options ranging from 0 (not at all) to 4 (very strong). Responses are averaged together, with higher scores reflecting greater BPD symptomatology. Examples of scale items include “My mood rapidly cycled in terms of anxiety, anger, and depression”, “Criticism had a devastating effect on me”, and “I felt disgusted by myself”. The authors reported a Cronbach’s α of 0.97 and strong evidence of convergent and discriminant validity. The Cronbach’s α of the BSL for the current study was 0.95.

#### 2.2.2. Difficulties in Emotion Regulation Scale

The Difficulties in Emotion Regulation Scale (DERS) [26] is a 36-item, Likert-type scale ranging from 1 (almost never) to 5 (almost always). Responses are averaged after reverse scoring select items, and higher scores indicate increased difficulty with emotional regulation. The scale is comprised of 6 factors, labeled Nonacceptance of Emotional Responses, Difficulties Engaging in Goal-Directed Behavior, Impulse Control Difficulties, Lack of Emotional Awareness, Limited Access to Emotion Regulation Strategies, and Lack of Emotional Clarity. Questions include: “I experience my emotions as overwhelming and out of control”, “When I’m upset, I become angry with myself for feeling that way”, and “When I’m upset, I have difficulty controlling my behaviors”. The authors reported a Cronbach’s α of 0.93, and construct validity was supported in the validation sample. For the current study, the DERS Cronbach’s α was 0.95 for the total scale. Subscale Cronbach’s α’s were as follows: Nonacceptance of Emotional Responses α = 94, Difficulties Engaging in Goal-Directed Behavior α = 0.89, Impulse Control Difficulties α = 0.90, Lack of Emotional Awareness α = 0.63, Limited Access to Emotion Regulation Strategies α = 0.90, and Lack of Emotional Clarity α = 0.84.

#### 2.2.3. Suicidal Behavior and Body Modification

The Suicidal Behavior and Body Damage and Modifications Scale (SBBDM-S) [4] evaluates the lifetime presence of suicidal behavior, NSSI, and body modifications. The scale consists of 10 items, with 1–4 investigating suicidal behavior, 5–8 investigating body modifications, and 9–10 investigating NSSI. Within the body modifications portion, three sub-scores indicated the total number of piercings, the percentage of tattoo body coverage, and the total body modifications score (sum of the previous two subscales). The SBBDM-S was adapted from its original form for the purpose of the current study. This adapted version consists only of the body modifications portion. Some additional questions were added to investigate the number of body modifications an individual has, their locations, and how long it has been since obtaining their most recent body modifications.

#### 2.2.4. Craving Experience Questionnaire

The Craving Experience Questionnaire (CEQ) [27] is a 10-item scale originally designed to assess the urgency of cravings in smokers and nicotine users, with response options on a spectrum from 0 (not at all) to 10 (extremely). Example items include “At that time, how strong was the urge to have …?” and “At that time, how vividly did you imagine how your body would feel?” and “At that time, how intrusive were the thoughts?” Crobach’s α was 0.91 for the strength form of the scale and 0.94 for the frequency form of the scale in the validation sample. The CEQ’s validity was supported among smokers and in a modified version assessing alcohol cravings. The current study adapted this scale to investigate cravings for NSSI and body modifications, removing two items asking about sensory stimuli associated with nicotine use. Cronbach’s α for the CEQ in the current study was 0.95 for the NSSI adaptation and 0.90 for the body modifications adaptation.

#### 2.2.5. Motivations for Body Modification Scale (MBMS)

Currently, no measures exist that assess a wide range of motivations for obtaining body modifications, especially pertaining to emotional regulation and self-expression. In a literature review, Wohlrab et al. [14] reported important motivations for body modifications such as beauty/art/fashion, individuality, personal narrative, physical endurance against pain, group affiliations/commitment, social resistance/protest, spirituality/cultural tradition, addiction to body modifications, sexual motivation, and no specific reason. Thus, we developed the MBMS to include items assessing these categories, along with emotional regulation and self-expression motivations for obtaining body modifications. The original MBMS consisted of 17 questions on a Likert-type scale ranging from 1 (strongly disagree) to 5 (strongly agree). Information regarding the development and initial validity of the scale and its factor structure are provided in Appendix A. The final MBMS included 15 total items, comprised of three factors assessing Emotion Regulation (α = 0.89), Expression/Autonomy (α = 0.81), and Social Identity (α = 0.63).

### 2.3. Procedure

This study was preregistered at https://osf.io/x5wkb/?view_only=f1dae1f78c3a4bf4b2d158fb60d12d1a (accessed on 9 December 2024), and responses were collected via Qualtrics. After consenting to the survey, participants completed questionnaires consisting of the BSL and the modified CEQ-NSSI. Participants were then asked if they had any body modifications (tattoos and piercings excluding first earlobe piercings). Participants who endorsed body modifications were asked to complete the body modification portion of the adapted SBBDM-S, the body modifications adaptation of the CEQ, and the MBMS (designed for this study). After completing the study, all participants were provided with a debriefing statement that further detailed the purpose of the study and contained a link to enter the raffle.

### 2.4. Statistical Analyses

We planned two linear regression analyses to evaluate (1) if the number of tattoos and piercings participants reported and tattoo coverage (predictor variables, entered simultaneously) accounted for variance in BSL scores (outcome variable); (2) if emotional regulation and self-expression (MBMS Emotion Regulation and Expression/Autonomy subscale scores, predictor variables entered simultaneously) accounted for variance in BSL scores (outcome variable); and (3) if NSSI craving scores significantly positively correlated with body modification craving scores. Correlations and descriptive statistics are also reported. Data and analyses were conducted using SPSS software version 28. Using G*Power version 3.1.9.5, sensitivity analysis showed that for a sample of 199, with a power of 0.90, with three tested predictors of seven total possible predictors, the analysis would have the power to detect an effect size of f^2^ = 0.073, which represents a small-to-medium effect size, suggesting sufficient sample size for the planned regression analyses in the current study.

## 3. Results

Under half of our sample (40.5%) reported having tattoos. Of those who reported tattoos, a majority (64.2%) reported four or fewer tattoos. Over half of our sample (55%) did not have piercings. Of those who have piercings, half (50%) of the sample reported having less than seven piercings. Thus, our sample had more tattoos than a large Pew research sample of American participants (15–24%) and more piercings than a national probability sample of 500 adults aged 18–50 (14%) [28,29]. Descriptive information and inter-correlations are provided in Table 2.

### 3.1. Body Modification and NSSI Craving: H1

Supporting H1, NSSI craving was strongly positively correlated with body modification craving among participants who had body modifications (*r* = 0.56, *p* < 0.001). See Table 2.

### 3.2. Extent of Body Modifications: H2

The model was statistically significant: the number and extent of tattoos and piercings explained a significant proportion of variance in BSL scores, F (3, 59) = 4.04, *p* = 0.011, *R*^2^ = 0.17. In contrast to hypotheses, the number of tattoos an individual had, along with the approximate percentage of tattoo body coverage, did not significantly account for the variance in BSL scores (*p* = 0.32 and *p* = 0.80, respectively). Supporting hypotheses, BSL score variance was significantly accounted for by the number of piercings an individual had (*p* = 0.001), such that each additional piercing was associated with a 0.10 increase in BSL score. See Table 3 for results.

### 3.3. Motivations for Body Modifications: H3

The model was not statistically significant, F (3, 129) = 1.36, *p* = 0.26, *R*^2^ = 0.03. Emotion Regulation, Expression/Autonomy, and Social Identity scores were not significantly associated with BSL scores (*p* = 0.16–0.72), contrary to hypotheses. There was some evidence of heteroscedasticity; however, transformations of the variables yielded similar non-significant results. Thus, untransformed data are reported for ease of interpretation. See Table 4 for results.

## 4. Discussion

The purpose of this study was to examine the relationship between BPD symptomatology, body modification, and motivations to acquire additional body modifications. This included assessing whether BPD symptoms were associated with body modifications, whether body modifications are motivated by emotional regulation and self-expression, and whether cravings to engage in NSSI were correlated with cravings to obtain additional body modifications. Blay et al. [4] investigated emotional regulation as a motivator for body modifications in a clinical population with BPD, finding that NSSI correlated with the total number of body modifications obtained by this population. The current study extends these results by looking at a non-clinical population, investigating self-expression motivations for body modifications, and examining the relationship between cravings for NSSI and body modification.

We hypothesized that the extent of body modifications an individual had would be associated with their level of BPD symptomatology, such that an increased amount of body modification would indicate greater BPD psychopathology. This had been found previously [4,18]. Contrary to our initial hypotheses, neither the number of tattoos nor the percentage of body coverage of tattoos was associated with BPD symptoms. However, the number of piercings an individual had did significantly explain variance in BPD symptomatology, as hypothesized. These results are consistent with findings from previous studies, in which individuals who reported engaging in NSSI, often related to BPD, had a greater number of piercings [4,22]. However, the lack of correlation between BPD features and tattoos is partly inconsistent with Blay et al. [4]. This may be due to our non-clinical sample, as Blay et al. collected data directly from an outpatient sample with BPD [4]. Blay and colleagues found significant associations between body modifications and total BPD scores, as measured by the Structured Clinical Interview for the DSM-5, but a non-significant association was found between body modification and BSL scores [4], which was the measure used in the current study. Thus, it is possible that the BSL is less sensitive to BPD symptoms that are associated with body modification than the Structured Clinical Interview for the DSM-5. Importantly, Blay et al. found that the number of piercings was significantly associated with emotional dysregulation, whereas the percentage of the body covered by tattoos was significantly associated with sensation-seeking [4]. Similarly, recent research among 762 non-clinical adults found a stronger relationship between piercings and symptoms of attention-deficit hyperactivity disorder than between tattoos and attention-deficit hyperactivity disorder symptoms, though both were statistically significant [30].

It is possible that having tattoos may be regarded as more socially acceptable than having piercings, especially when concerning more ‘extreme’ piercings [28]. This is supported by the rise in tattoo popularity, with an online Ipsos poll [31] reporting a rise in tattoos from 21% of the population in 2012 to 30% in 2019. Minimal research has been conducted on societal acceptance of individuals with multiple tattoos versus multiple piercings. However, based on the rise in tattoo acceptance broadly, individuals with and without BPD symptoms may be equally likely to get tattoos due to increased social acceptance, such that they may be less predictive of BPD psychopathology.

We hypothesized that participants with body modifications would be more likely to report emotional regulation and self-expression as motivations for obtaining their body modifications if they were higher in BPD pathology. In contrast to the hypotheses, neither emotional regulation nor self-expression motivations were significantly associated with BPD pathology. Emotional dysregulation is heightened in individuals who engage in NSSI, and NSSI appears to serve as an emotional regulator [32]. If NSSI and body modifications are interconnected among individuals high in BPD symptomatology, then emotional regulation may serve as a motivational factor for obtaining body modifications, similar to its motivating NSSI. Although emotional regulation motivations for body modifications were not significantly associated with BPD psychopathology, as hypothesized, they were significantly positively correlated with NSSI cravings (*r* = 0.24, *p* < 0.001), body modification craving (*r* = 0.49, *p* < 0.001), and the percent of body coverage by tattoos (*r* = 0.29, *p* < 0.001), the number of tattoos (*r* = 0.30, *p* < 0.001), and the number of piercings (*r* = 0.24, *p* < 0.001), strongly suggesting the interrelationships of these motivations for body modification and both craving for NSSI and further body modifications (see Table 2). Additionally, although self-expression/autonomy motivations for body modifications were not significantly associated with BPD symptoms, as hypothesized, they were significantly positively correlated with body modification cravings (*r* = 0.5, *p* < 0.001), and the number of tattoos (*r* = 0.23, *p* < 0.05) and piercings (*r* = 0.21, *p* < 0.05).

As hypothesized, NSSI craving scores were significantly associated with body modification craving scores. Importantly, both NSSI and body modification cravings were significantly correlated with almost all DERS subscales as well, even though the DERS was not associated with the number and percentage coverage of body modifications. Given that emotional regulation motivations for body modifications were significantly associated with NSSI craving scores, people may use body modifications as an alternative form of emotional regulation that is more socially acceptable than engaging in self-harming behaviors. This is supported by case studies, including the young man who reported using the pain of tattooing as a distraction from NSSI cravings and emotional fluctuations [24]. Similarly, a veteran who met the criteria for multiple psychiatric disorders, including BPD, disclosed that he had engaged in typically observed NSSI behaviors but preferred the pain of tattooing due to its social acceptance. He reported that the tattooing process combatted negative emotional states, tension, and feelings of numbness [33,34]. The connection between NSSI and body modification may, therefore, be driven by emotional regulation for individuals with BPD symptoms, with body modifications being endorsed by some individuals due to their greater social acceptance over other NSSI methods. However, some people with BPD may not view body modification as a means of regulating their emotions, but emotional regulation may be a side effect of body modification. This is likely due to the endorphin release from self-injury: Directly after engaging in NSSI, individuals demonstrated a significant increase in salivary β-endorphin levels [35]. These endorphin releases have not been explicitly studied with regard to body modifications, although Wohlrab et al. [14] and Weiler et al. [36] interpolate that this may be a motivation for body modifications among some individuals.

It is likely that many individuals without significant psychopathology are motivated to obtain additional body modifications as a means of self-expression and autonomy and for social identity reasons, as our scale suggested. Additional research is needed to better understand under what circumstances and for whom differing motivations may guide decisions to obtain body modifications.

### 4.1. Limitations

Despite the important findings in the current study, there are some notable limitations. First, the study was cross-sectional in nature; our data is therefore only correlational and cannot be used to infer causality. Second, there is also a potential for self-report and selection bias despite efforts to recruit a diverse sample. Third, demographic information was missing for half of the participants due to researcher error. This diminishes the ability to draw conclusions regarding differences between demographic groups, rendering the generalizability of the current findings unknown. Fourth, although the study was advertised in multiple locations to attract a range of respondents, many participants were younger adults, based on the available demographic information, who may have only recently become old enough to legally obtain body modifications without parental consent. Additionally, we did not question participants regarding cultural–historical motivations for body modifications. Two recent reviews of motivations for body modifications do speak in greater detail about cultural, historical, and social motivations for body modifications [36,37]. Furthermore, we did not account for recruitment sources in our survey and are therefore unable to provide a breakdown of how many participants found the study from various recruitment sources and are therefore unable to with confidence confirm heterogeneity in our sample beyond the existing demographic information. Additionally, the emotional regulation and self-expression motivations for body modification scales were developed solely for this study and were not previously validated measures. The Social Identity subscale was brief, consisting of only three items, and internal consistency was inadequate. We only measured NSSI craving using a scale initially designed to measure substance use craving, though some evidence suggests these cravings differ from one another [8]. While the BSL scale had many questions on thoughts of NSSI, there were no subscales examining the incidence of NSSI itself. These omissions, while limitations, were in pursuit of survey brevity, as shorter surveys have been shown to increase survey completion [38]. Finally, we did not collect longitudinal data on the progression of body modifications along with BPD symptoms and NSSI and body modification craving over time.

### 4.2. Future Directions

Future researchers should survey a larger group of participants with more body modifications so that participants have a wide range of body modification coverage for analyses. Additionally, researchers can determine if age is a factor in the relationship between BPD symptoms and body modification. Body modifications may be a more modern form of coping or self-expression endorsed by younger individuals, whereas older adults may have developed more generationally sanctioned coping skills or other adaptive coping skills over time. This is supported by the finding that the severity of BPD symptoms wanes with age [39]. Future research should, as previously mentioned, examine endorphin release and body modifications. Additionally, future research should survey people with BPD and investigate if their level of concern with social acceptability moderates their preference for either NSSI or body modification. Individuals with BPD who are seeking a more socially acceptable method of inflicting pain on themselves to regulate their emotions may elect to obtain body modifications, which are more likely to be perceived as “cool” rather than living with self-harm scars that may result in stigma [40]. Lastly, longitudinal and experience-sampling methods may better identify the temporal relationships between NSSI craving, body modification craving, and engaging in NSSI and body modification.

### 4.3. Conclusions and Implications

The implications of this study are important for furthering understanding of BPD and related psychopathology. Clinicians may benefit from assessing whether a patient is using body modification as an NSSI alternative and if it is adaptive or maladaptive. Overall, there are many research gaps involving body modifications and their possible emotion regulation function, considering their recent rise in popularity.

## Figures and Tables

**Table 1 ijerph-22-00089-t001:** Demographic Information on Gender, Race, Ethnicity, and Household Income of Participants.

Category	Subcategory	Percent
Gender	Cisgender women	35.0%
	Cisgender men	12.5%
	Transgender woman	0.0%
	Transgender man	0.0%
	Nonbinary	3.5%
	Prefer not to say	1.0%
	Data not collected	48.0%
Race	White or Caucasian	40.0%
	Black or African American	2.0%
	Native American	0.5%
	Asian	2.5%
	Native Hawaiian or Pacific Islander	0.0%
	Mixed Race	4.5%
	Other	1.5%
	Prefer not to say	3.0%
	Data not collected	46.0%
Ethnicity	Spanish/Latino/Hispanic origin	4.5%
	Not of Spanish/Latino/Hispanic origin	46.5%
	Prefer not to say	1.0%
	Data not collected	48%
Household Income	Less than USD 25,000	6.0%
	USD 25,000–USD 49,999	6.0%
	USD 50,000–USD 74,999	7.0%
	USD 75,000–USD 99,999	5.0%
	USD 100,000–USD 149,999	6.5%
	USD 150,000 or more	9%
	Unsure	8.5%
	Prefer not to say	4.0%
	Data not collected	48.0%

**Table 2 ijerph-22-00089-t002:** Correlation Matrix for the Motivation for Body Modification Scale (MBMS) and Related Measures.

	1	2	3	4	5	6	7	8	9	10	11	12	13	14	15	16
1. MBMS—Emotional Regulation	1.0															
2. MBMS—Expression/Autonomy	0.58 ***	1.0														
3. MBMS—Social Identity	0.23 **	0.22 *	1.0													
4. Borderline Symptom List	0.17	0.15	0.06	1.0												
5. DERS Nonacceptance	0.10	0.07	−0.02	0.65 ***	1.0											
6. DERS Goal-directed Beh.	0.11	0.21 *	−0.13	0.59 ***	0.53 ***	1.0										
7. DERS Impulse control	0.16	0.11	0.03	0.65 ***	0.64 ***	0.56 ***	1.0									
8. DERS Awareness	0.01	−0.11	0.09	0.27 ***	0.39 ***	0.08	0.19 **	1.0								
9. DERS Strategies	0.10	0.04	−0.04	0.76 ***	0.70 ***	0.62 ***	0.74 ***	0.27 ***	1.0							
10 DERS Clarity	0.07	−0.02	0.00	0.50 ***	0.55 ***	0.32 ***	0.44 ***	0.64 ***	0.54 ***	1.0						
11. DERS Global	0.13	0.07	−0.02	0.77 ***	0.86 ***	0.70 ***	0.81 ***	0.51 ***	0.89 ***	0.73 ***	1.0					
12. NSSI Craving	0.24 **	0.15	−0.09	0.65 ***	0.52 ***	0.37 ***	0.51 ***	0.21 **	0.55 ***	0.36 ***	0.57 **	1.0				
13. Body Modification Craving	0.49 **	0.55 **	0.03	0.44 **	0.30 **	0.28 **	0.36 **	0.08	0.31 **	0.23 **	0.35 **	0.56 **	1.0			
14. Percent Tattoos	0.29 **	0.19	0.37 **	−0.10	−0.15	−0.16	−0.003	0.006	−0.01	−0.13	−0.09	0.07	0.23 *	1.0		
15. Number Tattoos	030 **	0.23 *	0.41 **	−0.07	−0.14	−0.15	−0.05	0.01	−0.05	−0.12	0.10	0.23 *	0.23 *	0.81 **	1.0	
16. Number Piercings	0.26 **	0.21 *	−0.07	0.20 *	0.10	0.06	0.09	0.14	0.10	0.16	0.15	0.20 *	0.30 **	0.03	0.20	1.0
*M*	3.19	3.74	2.03	1.93	2.31	3.10	1.99	2.63	2.31	2.35	2.43	1.33	2.74	1.26	4.77	3.44
*SD*	0.90	0.88	0.94	0.73	1.07	0.98	0.88	0.69	0.91	0.82	0.69	2.13	2.26	0.49	6.03	2.36
*N*	126	126	126	199	199	199	199	199	199	199	199	199	126	81	81	110

Note. MBMS = Motivations for Body Modification; DERS = Difficulties in Emotion Regulation Scale; Nonacceptance = Nonacceptance of emotional responses; Goal-directed Beh. = Difficulty engaging in goal-directed behavior; Impulse = impulse control difficulties; Awareness = lack of emotional awareness; Strategies = Limited access to emotion regulation strategies; Clarity = Lack of emotional clarity; NSSI = Non-suicidal self-injury; * *p* < 0.05; ** *p* < 0.01; *** *p* < 0.001.

**Table 3 ijerph-22-00089-t003:** Multiple Linear Regression of Extent of Body Modification Variables on Borderline Symptom List Scores.

Variables	*B*	*SE*	*β*	*p*	Part *r*
Number of tattoos	−0.03	0.03	−0.20	0.32	−0.12
Percentage of tattoo body coverage	0.07	0.28	0.05	0.80	0.03
Number of piercings **	0.10	0.03	0.42	0.001	0.40

Note: Variables were analyzed in a multiple linear regression with borderline symptom list scores as the dependent variable. ** *p* < 0.01

**Table 4 ijerph-22-00089-t004:** Linear Regression Analyses of Body Modification Motivations on Borderline Symptom List Scores.

Dependent Variable	Independent Variable	*B*	*SE*	*β*	*p*	Part *r*
BSL score	Emotion Regulation Motivation	0.10	0.093	0.12	0.29	0.10
	Expression/Autonomy Motivation	0.07	0.01	0.08	0.72	0.064
	Social Identity Motivation	0.01	0.07	0.02	0.16	0.014

Note. BSL = Borderline Symptom List.

## Data Availability

This study was preregistered here: https://osf.io/x5wkb/?view_only=f1dae1f78c3a4bf4b2d158fb60d12d1a (accessed on 9 December 2024), and data will be made publicly available following the publication of the manuscript.

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
