# Peer review of "Borderline Personality Symptoms, Body Modification, and Emotional Regulation"

_ijerph, 2025, doi:10.3390/ijerph22010089_

Round 1
Reviewer 1 Report (Previous Reviewer 2)
Comments and Suggestions for Authors
The authors have completely addressed all my comments, and I have no further concerns. Therefore, I recommend accepting the paper.
Comments on the Quality of English LanguageThe English could be improved to more clearly express the research.
Reviewer 2 Report (Previous Reviewer 1)
Comments and Suggestions for Authors
Thank you
This manuscript is a resubmission of an earlier submission. The following is a list of the peer review reports and author responses from that submission.
Round 1
Reviewer 1 Report
Comments and Suggestions for Authors
This paper would have great potential, but there are many limitations that turn off enthusiasm. I would point out some urgent changes that need to be made to the manuscript, but I am of the opinion that it does not meet the requirements for publication in this journal. And I am very sorry about this. In my opinion, the research should be repeated, or at least expanded, enriched in addition with a qualitative part and other quantitative measures (such as ACEs).
- the abstract should be modified: reduce the theoretical explanation and include information about the sample (sociodemographic data and size), the experimental design used, and the nationality of the study.
- there are some terms, such as NSSI craving or body modification craving that should be better explained, as I do not know of the existence of an adequate rational that signals the presence of craving in these activities. Perhaps it should be better clarified what the authors meant.
- It seems to me that the sample is very small indeed. A survey of a normative population from which borderline traits are detected should be larger.
- The representativeness of the sample is not there. It is not very clear how it was selected, and offering a reward I think is not an appropriate strategy for this type of population. I am also not convinced by the age range: perhaps there is some cultural-historical influence on the use of tattoos and body modifications. Perhaps a larger sample focused on a specific age range might be more useful.
- There are also errors in the collection of demographic data. This is really limited, and in my opinion the research would deserve more attention.
- The discussion is very poor, the limitations of the research, future directions, and practical implications should be better developed and explored.
Author Response
This paper would have great potential, but there are many limitations that turn off enthusiasm. I would point out some urgent changes that need to be made to the manuscript, but I am of the opinion that it does not meet the requirements for publication in this journal. And I am very sorry about this. In my opinion, the research should be repeated, or at least expanded, enriched in addition with a qualitative part and other quantitative measures (such as ACEs).
- the abstract should be modified: reduce the theoretical explanation and include information about the sample (sociodemographic data and size), the experimental design used, and the nationality of the study.
Thank you for this feedback. We have edited the abstract to include more sample information, as well as information on our experimental design and where our study was conducted.
- there are some terms, such as NSSI craving or body modification craving that should be better explained, as I do not know of the existence of an adequate rational that signals the presence of craving in these activities. Perhaps it should be better clarified what the authors meant.
Thank you very much for this feedback. We did not include a direct measure of NSSI incidence due to our non-clinical sample, and for survey brevity. Because our hypotheses only included craving to engage in NSSI, we chose to only assess this construct. Many patients report NSSI cravings in clinical settings, and we have cited two studies on this in the introduction when defining NSSI . That being said, these are limitations of the study. We have addressed all of the above in the discussion sections limitations section.
“We only measured NSSI craving using a scale initially designed to measure substance use craving, though some evidence suggests these cravings differ from one another [7]. While the BSL scale had many questions on thoughts of NSSI, there were no subscales examining the incidence of NSSI itself.These omissions, while limitations, were in pursuit of survey brevity – as shorter surveys have been shown to increase survey completion [31].”
- It seems to me that the sample is very small indeed. A survey of a normative population from which borderline traits are detected should be larger.
We appreciate the Reviewer’s attention to detail and agree, so have provided a sensitivity analysis for the data, as follows:
“G*Power sensitivity analysis showed that for a sample of 199, with power of .90, with three tested predictors of seven total possible predictors, the analysis would have power to detect an effect size of f2 = .073, which represents a small-to-medium effect size, suggesting sufficient sample size for the planned regression analyses were present in the current study.”
- The representativeness of the sample is not there. It is not very clear how it was selected, and offering a reward I think is not an appropriate strategy for this type of population. I am also not convinced by the age range: perhaps there is some cultural-historical influence on the use of tattoos and body modifications. Perhaps a larger sample focused on a specific age range might be more useful.
Thank you very much for this feedback. We selected our sample through a combination of methods; our departmental subject pool, flyers distributed across college campus, posted on social media, and posted in local tattoo parlors and hair salons in a mid-sized metropolitan area in the Northeast U.S. to survey a wide variety of individuals - as clarified in our “participants” section. Our sample was non-clinical, and not all participants were rewarded - as also clarified in the participants section. We have added the comment on cultural-historical influence on the use of tattoos and body modifications to our limitations.
“First, the study was cross-sectional in nature; our data is therefore only correlational and cannot be used to infer causality. There is also a potential for self-report and selection bias despite efforts to recruit a diverse sample.”
- There are also errors in the collection of demographic data. This is really limited, and in my opinion the research would deserve more attention.
This is correct. We were very transparent about this error, and have noted it as a limitation in multiple places throughout our manuscript. We did collect demographic information for 95 of our participants. Again, we regret the omission of demographic information for half of the sample; however, we note that it is not atypical for there to be some researcher error within many studies, but that it does not necessarily preclude the utility of publishing that research, with the understanding that future research will need to build on that, by replicating and extending the research.
- The discussion is very poor, the limitations of the research, future directions, and practical implications should be better developed and explored.
Thank you for your comment. Based on yours and other Reviewers’ feedback, we have improved our discussion by adding to our limitations, future directions, and conclusions and implications sections and labeling the sections of our discussion for organization and clarity purposes.
Reviewer 2 Report
Comments and Suggestions for Authors
- Abstract should be rewritten for better clarity and focus on key points.
- The description of Borderline Personality Disorder (BPD) should be more detailed.
- Add a section on study limitations and future research directions.
- Incorporate additional psychometric tests to strengthen the methods.
- Include a G*Power analysis to justify the sample size selection.
- Present a comparison of findings with existing literature in tabular form.
- Add a table listing all abbreviations used in the manuscript for clarity.
Abstract: The abstract lacks focus and does not effectively communicate the study's core objectives, methodology, or implications. A clearer, more concise summary that highlights the research hypothesis, methods, key findings, and implications is needed. Without this, the reader is left with an incomplete understanding of the study's relevance.
Introduction: While the introduction provides some background, it fails to deliver a comprehensive understanding of Borderline Personality Disorder (BPD) and its connection to emotional regulation and body modification. The lack of depth in discussing BPD symptoms weakens the rationale behind the study. A thorough literature review that explores BPD’s diagnostic criteria, emotional dysregulation, and motivations for non-suicidal self-injury (NSSI) and body modification is essential to strengthen the manuscript.
Methodology (Page 5, Lines 12-18): The methods section raises significant concerns. The study relies heavily on a narrow selection of psychometric tests (BSL and DERS), without sufficient justification for why these tools were chosen over more comprehensive measures. Furthermore, the absence of a G*Power analysis to justify the sample size is a glaring omission. The lack of a sample size calculation weakens the validity of the findings, as readers are unable to determine if the study is appropriately powered. A stronger methodological foundation is needed to ensure the reliability of the results.
Participants: The manuscript lacks clarity on how the sample size was determined. There is no discussion on how potential biases were controlled during participant recruitment. Although participants were recruited from various locations, the authors fail to address how this could introduce variability and affect the generalizability of the results.
Statistical Analysis (Page 7, Lines 20-25): The statistical methods are not appropriately justified. The selection of linear regression seems inadequate given the complexity of the relationship between BPD symptoms and body modification. More sophisticated statistical models, such as mediation or moderation analysis, should be considered to capture the full depth of the research question. Relying on simple regression oversimplifies the study’s hypotheses and weakens the potential insights into these psychological constructs.
Results & Interpretation: The results section is underdeveloped. While statistical analyses are presented, there is little interpretation of the data in the context of broader psychological theories or prior research. The manuscript makes intriguing claims, such as “piercings predict BPD symptomatology more strongly than tattoos” (Page 9, Lines 5-10), but does not provide sufficient theoretical or empirical backing for such statements. A comparative table with existing literature would help address these inconsistencies and clarify how this study contributes to the field.
Discussion & Limitations: The discussion fails to critically engage with the findings or reflect on their implications. Important limitations, such as the homogeneity of the sample, the potential for self-report bias, and the lack of longitudinal data, are not addressed. A critical evaluation of these factors is necessary to provide transparency and context for the study's conclusions. Additionally, the lack of a section on future research directions is a significant oversight. The authors should suggest areas for further investigation, which would contribute to the development of this important area of research.
Structural Issues: The manuscript suffers from organizational weaknesses. Key terms, such as abbreviations, are not defined early enough, which affects the readability and flow of the text. The addition of a table listing all abbreviations would greatly enhance clarity and help readers follow the complex terminology used throughout the paper.
Spelling/Grammar Error (Page 3, Line 14): The term "symptomology" is incorrectly used in the text. The correct term is "symptomatology." Attention to such details is crucial for maintaining the academic rigor expected in peer-reviewed publications.
Conclusion: Overall, the manuscript lacks the necessary methodological rigor and clarity to support its conclusions. The introduction needs a more thorough exploration of the theoretical underpinnings, while the methods section requires significant revision, particularly with the inclusion of more appropriate psychometric tools and a G*Power analysis. The statistical methods are too simplistic for the complexity of the research question, and the results are not sufficiently interpreted in light of existing literature. Major improvements in structure, depth of discussion, and transparency regarding limitations are essential before this manuscript can be considered for publication.
Minor editing of English language required.
Author Response
[Reviewer 2 had provided these summary bullet points. Each is addressed in greater detail below by Reviewer 2, and our responses]
Abstract should be rewritten for better clarity and focus on key points.
The description of Borderline Personality Disorder (BPD) should be more detailed.
Add a section on study limitations and future research directions.
Incorporate additional psychometric tests to strengthen the methods.
Include a G*Power analysis to justify the sample size selection.
Present a comparison of findings with existing literature in tabular form.
Add a table listing all abbreviations used in the manuscript for clarity.
Abstract: The abstract lacks focus and does not effectively communicate the study's core objectives, methodology, or implications. A clearer, more concise summary that highlights the research hypothesis, methods, key findings, and implications is needed. Without this, the reader is left with an incomplete understanding of the study's relevance.
Thank you for this feedback. We have rewritten the abstract for brevity and clarity.
Introduction: While the introduction provides some background, it fails to deliver a comprehensive understanding of Borderline Personality Disorder (BPD) and its connection to emotional regulation and body modification. The lack of depth in discussing BPD symptoms weakens the rationale behind the study. A thorough literature review that explores BPD’s diagnostic criteria, emotional dysregulation, and motivations for non-suicidal self-injury (NSSI) and body modification is essential to strengthen the manuscript.
Thank you for this feedback. We initially omitted a more thorough literature review of BPD for brevity purposes to meet journal word count limits, and due to our sample being non-clinical. However, we have added literature to provide more information on the relationship between borderline personality disorder, emotional regulation, and NSSI:
“About 1.6% of the global population is estimated to meet the diagnostic criteria for borderline personality disorder (BPD) [1]. The Diagnostic and Statistical Manual of Mental Disorders-5-TR (DSM-5-TR) defines BPD as “a pervasive pattern of instability of interpersonal relationships, self-image, and affects, and marked impulsivity, beginning by early adulthood and present in a variety of contexts” [1]. Diagnostic criteria for BPD include desperate efforts to avoid abandonment, volatile interpersonal relationships, identity disturbance, impulsivity, suicidal ideation and behavior, mood instability, feelings of chronic emptiness, intense anger, paranoid ideation, and non-suicidal self-injury (NSSI) [1]. In particular, BPD is a disorder characterized by emotional dysregulation, which may lead individuals to seek any release from overwhelming feelings, even if that release is maladaptive in the long-term. Clinicians are advised to look for cutting scars and other evidence of NSSI when evaluating a patient with BPD [3]. Treatments for BPD focus on attachment, mindfulness, emotional regulation, and interpersonal skills [4].”
However, there is very minimal research, thus far, on the relationship between borderline personality disorder pathology, emotional regulation, and body modification. All studies which have assessed this intersection have been discussed in the Introduction. Thus, we have also emphasized the lack of pre-existing literature and noted that this novel research area is a strength of the current study in the manuscript.
Methodology (Page 5, Lines 12-18): The methods section raises significant concerns. The study relies heavily on a narrow selection of psychometric tests (BSL and DERS), without sufficient justification for why these tools were chosen over more comprehensive measures.
We were keeping the survey brief in order to retain participants and to maximize participant quality, as shorter surveys are more likely to be completed (Galesic & Bosnjak, 2009). We have added this information to the manuscript. The DERS is one of the most commonly used measures of emotion regulation with 12,756 citations, according to Google Scholar, and we have added this to the Measures section. We have noted that the BSL also has evidence of reliability and validity from diverse samples across a number of studies to the Measures section.
Furthermore, the absence of a G*Power analysis to justify the sample size is a glaring omission. The lack of a sample size calculation weakens the validity of the findings, as readers are unable to determine if the study is appropriately powered. A stronger methodological foundation is needed to ensure the reliability of the results.
We appreciate the Reviewer’s attention to detail and agree, so have provided a sensitivity analysis for the data, as follows:
“G*Power sensitivity analysis showed that for a sample of 199, with power of .90, with three tested predictors of seven total possible predictors, the analysis would have power to detect an effect size of f2 = .073, which represents a small-to-medium effect size, suggesting sufficient sample size for the planned regression analyses were present in the current study.”
Participants: The manuscript lacks clarity on how the sample size was determined.
We appreciate this feedback. This began as a student project, and so the largest possible sample size was recruited with the time and monetary limitations we had available. We provided the sensitivity analysis, as noted above, to demonstrate that the sample size is sufficient to examine our planned analyses. We also added the following to the Participants section: “Sample size was determined by the time frame available for completion of this study by the internal funding mechanism, attempting to recruit as many participants as possible within that time frame for adequate statistical power. Responses were collected from October 2023 to January 2024”
There is no discussion on how potential biases were controlled during participant recruitment. Although participants were recruited from various locations, the authors fail to address how this could introduce variability and affect the generalizability of the results.
Thank you for this feedback. We added the following information to the Participants section: “This recruitment strategy was selected to maximize variability within the sample in terms of age, and presence/absence of body modifications, which should improve generalizability of the results. We were specifically targeting individuals who would have higher numbers of body modifications by recruiting from tattoo and piercing parlors in person and via their social media platforms, and recruited in hair salons to balance the sample with individuals who may be less likely to have body modifications.”
We addressed the possibility of selection bias in the limitations as well, as follows:
“There is also a potential for self-report and selection bias despite efforts to recruit a diverse sample.”
Statistical Analysis (Page 7, Lines 20-25): The statistical methods are not appropriately justified. The selection of linear regression seems inadequate given the complexity of the relationship between BPD symptoms and body modification. More sophisticated statistical models, such as mediation or moderation analysis, should be considered to capture the full depth of the research question. Relying on simple regression oversimplifies the study’s hypotheses and weakens the potential insights into these psychological constructs.
We appreciate the Reviewer’s suggestion and have considered it. However, there are a few reasons why mediational analyses are not optimal for the current study, after further consideration. We considered running three separate mediation analyses with multiple mediators, with X1 = Number of Tattoos, X2 = percentage of tattoo coverage, and X3 = Number of Piercings for the three mediation models. All three mediation models would include the three mediators: Emotional Regulation, Expression/Autonomy, and Social Identity, and the dependent variable of BSL scores (borderline personality disorder symptoms). However, three main reasons not to do this are (1) it was not our original hypothesis, as registered on osf.io and goes against principles of open science; (2) a post-hoc power analysis suggests that we would need very large samples in order to run these analyses such that it would not be feasible (I ran power analysis with up to 800 participants for one mediation model with the lowest correlation sizes between any X and our three mediators and Y variable, and the highest power for any mediation effects would have been .5); (3) our correlation table, which is now included in the main analysis rather than just in the supplement, per Reviewer 3’s suggestion, shows that before running mediation, not all of the X variables are significantly correlated with the mediator variables, not all the mediator variables are significantly correlated with the Y variable. Thus it does not meet the basic prerequisites one typically wants prior to running a mediation analysis (Baron & Kenny, 1986); and (4) there is not sufficient prior research in this area to suggest mediation, and mediation analyses or more complex analytical approaches should ideally be based on sound theoretical reasoning. Because this paper is one of the first to examine the relationships between these variables, correlation analyses, descriptive statistics, and simpler hypotheses that do have theoretical foundations may be more appropriate. Should the Reviewers and Editor still want these mediation analyses included, we are happy to add them to the supplement in a subsequent revision.
Results & Interpretation: The results section is underdeveloped. While statistical analyses are presented, there is little interpretation of the data in the context of broader psychological theories or prior research. The manuscript makes intriguing claims, such as “piercings predict BPD symptomatology more strongly than tattoos” (Page 9, Lines 5-10), but does not provide sufficient theoretical or empirical backing for such statements. A comparative table with existing literature would help address these inconsistencies and clarify how this study contributes to the field.
Thank you for this feedback. We have provided the following theoretical background in our discussion section: "However, the number of piercings an individual had did significantly predict BPD symptomatology, as hypothesized. These results are consistent with findings from previous studies, in which individuals who reported engaging in NSSI, often related to BPD, had a greater number of piercings [3, 17]."
Given the very limited research examining the relationships between BPD symptoms, NSSI, and body modifications, we felt that reviewing research in this area in the literature review and discussion provided sufficient detail about what is known in this area, without necessitating the addition of a table. However, if the Reviewer and Editor still would like to see a table, we are happy to add one in a subsequent revision. To put the current findings into greater contextwe have added three very recent articles to our discussion. Two are review describing the functions of body modification and one is an association of body modification to ADHD symptoms in a large nonclinical sample:
- Glans MR, Nilsson J and Bejerot S (2024) Tattoos, piercings, and symptoms of ADHD in non-clinical adults: a cross-sectional study. Front. Psychiatry 14:1224811. doi: 10.3389/fpsyt.2023.1224811
- Owens, R., Filoromo, S.J., Landgraf, L.A. et al. Deviance as an historical artefact: a scoping review of psychological studies of body modification. Humanit Soc Sci Commun 10, 33 (2023). https://doi.org/10.1057/s41599-023-01511-6
- Weiler, S. M., Mühlenbeck, C., & Jacobsen, T. (2024). Body alteration: On the mental function of body modification and body decoration. Culture & Psychology, 0(0). https://doi.org/10.1177/1354067X241242414
Discussion & Limitations: The discussion fails to critically engage with the findings or reflect on their implications. Important limitations, such as the homogeneity of the sample, the potential for self-report bias, and the lack of longitudinal data, are not addressed. A critical evaluation of these factors is necessary to provide transparency and context for the study's conclusions. Additionally, the lack of a section on future research directions is a significant oversight. The authors should suggest areas for further investigation, which would contribute to the development of this important area of research.
Thank you for your feedback. We have addressed the homogeneity of the sample, potential for self-report bias, and lack of longitudinal data in the limitations section - along with other areas suggested by reviewer 3: “First, the study was cross-sectional in nature; our data is therefore only correlational and cannot be used to infer causality. There is also a potential for self-report and selection bias despite efforts to recruit a diverse sample.” We have also labeled the sections of our discussion for clarity and organization.
Structural Issues: The manuscript suffers from organizational weaknesses. Key terms, such as abbreviations, are not defined early enough, which affects the readability and flow of the text. The addition of a table listing all abbreviations would greatly enhance clarity and help readers follow the complex terminology used throughout the paper.
Thank you for your feedback. We have added a list of all abbreviations used in the manuscript following the keywords.
Spelling/Grammar Error (Page 3, Line 14): The term "symptomology" is incorrectly used in the text. The correct term is "symptomatology." Attention to such details is crucial for maintaining the academic rigor expected in peer-reviewed publications
Thank you for this important feedback, we have corrected all instances of this miswording - a significant oversight on our part.
Conclusion: Overall, the manuscript lacks the necessary methodological rigor and clarity to support its conclusions. The introduction needs a more thorough exploration of the theoretical underpinnings, while the methods section requires significant revision, particularly with the inclusion of more appropriate psychometric tools and a G*Power analysis. The statistical methods are too simplistic for the complexity of the research question, and the results are not sufficiently interpreted in light of existing literature. Major improvements in structure, depth of discussion, and transparency regarding limitations are essential before this manuscript can be considered for publication.
We appreciate the Reviewer’s feedback and believe that the revision improves on these areas of concern and has strengthened the manuscript as a result.